# The Effects of Psychophysical Methods on Spectral and Spatial TOJ Thresholds

**DOI:** 10.3390/s22134830

**Published:** 2022-06-26

**Authors:** Leah Fostick, Harvey Babkoff

**Affiliations:** 1Department of Communication Disorders, Ariel University, Ariel 40700, Israel; 2Department of Psychology, Bar-Ilan University, Ramat-Gan 5290002, Israel; babkofh@gmail.com

**Keywords:** spatial temporal order judgment (TOJ), spectral TOJ, adaptive procedure, method of constant stimuli

## Abstract

(1) Background: A large number of studies have used different psychophysical methods for measuring temporal order judgment (TOJ) thresholds, which makes it difficult to compare the results of different studies. In this study, we aimed to compare the thresholds measured by the two main procedures used in many studies, the adaptive procedure, and the method of constant stimuli; (2) Methods: Study 1 tested spatial TOJ and included 109 participants, 50 using the adaptive procedure and 59 using the constant stimuli procedure. Study 2 tested spectral TOJ and included 223 participants, 119 using the adaptive procedure and 104 using constant stimuli; (3) Results: Both the spatial and spectral TOJ results showed no difference between the psychophysical methods, either in (1) the form of the distribution; (2) the mean; or (3) the standard deviation. However, Bayesian analysis showed a large Bayes factor only for spatial TOJ; (4) Conclusions: There is no difference between spatial TOJ thresholds measured by an adaptive procedure and the method of constant stimuli, and their results can be compared across studies. A similar conclusion can be drawn also for spectral TOJ, but should be considered more cautiously.

## 1. Introduction

Auditory temporal order judgment (TOJ) refers to the individual’s ability to correctly perceive the temporal order of at least two auditory stimuli [1,2,3,4,5,6,7], and has been used to compare the performance of control participants to sub-populations, such as dyslexic readers [8,9,10,11,12,13,14], aging adults [1,15,16,17,18], sleep deprived young adults [19,20], participants with attention deficit hyperactivity disorder (ADHD) [21], and aphasic patients [22,23].

TOJ thresholds are defined as the inter-stimulus interval (ISI) (or stimulus-onset-asynchrony, SOA) required for judging the order of two stimuli. One of the difficulties with comparing TOJ thresholds for different sub-populations is that the thresholds reported for the control participants (young healthy adults) vary considerably across studies (Table 1). The broad range of TOJ thresholds may be explained by several factors: (1) the use of different stimuli (broad band clicks or pure tones) by the different researchers; (2) the use of different stimulus durations (5–40 ms duration for tones and 1 ms for clicks); and (3) the use of different methodologies. Previously, we showed that stimulus duration, stimulus frequency, and type of stimulus affect TOJ thresholds [1,4,24]. For example, shortening the tone duration in the spatial TOJ paradigm resulted in an increase in the ISI threshold equivalent to the decrease in tone duration. In addition, the spectral TOJ thresholds (two pure tones differing in their frequency presented to the same ear(s)) always yielded shorter thresholds than the spatial TOJ thresholds (measured by the presentation of the same frequency tone asynchronously to the two ears). The present study was designed to test the effect of two different psychophysical methods on spatial and spectral TOJ thresholds.

The measurement of participants’ perceptual ability may be determined using psychophysical methods in which the values of the stimuli change gradually according to the participants’ responses, or using methods in which the values of the stimuli are decided before the experiment and presented randomly to the participant. The former method, the adaptive procedure, is subject to bias related to the habituation of participants to previously presented stimuli or to expectations about future stimuli. The latter procedure, the method of constant stimuli, can overcome these biases, but it is time-consuming, and requires many trials for reliable determination of the threshold.

Which psychophysical method is more valid or more reliable for determining thresholds? Several studies have addressed this question using different psychophysical tasks. Some studies compared the thresholds obtained by each method in an attempt to discover which method better reflects the individual’s ability [33,34], while others focused on their effectiveness in an attempt to obtain the most accurate measure that requires the least time and effort by the participants [35,36]. However, no study to date has tested whether the adaptive procedure and the method of constant stimuli produce the same thresholds in spectral and spatial TOJ. Thus, it is not clear whether it is possible to draw conclusions based on data collected via both methods. Moreover, in the era of big data based on analyses from different datasets, it is important to know whether thresholds collected using different methods can be combined. The aim of the present study was to test the effects of two different psychophysical methods on the two types of auditory temporal processing paradigms, spatial TOJ (Study 1) and spectral TOJ (Study 2). Since the adaptive procedure and the method of constant stimuli are used by a large number of researchers for measuring TOJ thresholds, we chose to compare the effects of these two methodologies on spatial and spectral TOJ thresholds. It should be noted that thresholds for the method of constant stimuli are calculated as the ISI for 75% correct responses, while thresholds in the adaptive procedure are equivalent to 70.7% correct responses. Since the aim of the current study was to facilitate comparisons between data reported in the literature, the main analyses used these threshold measures (and thus, comparisons between thresholds obtained for 75% and 70.7% correct responses, respectively). Nevertheless, we also added a direct comparison of the adaptive and constant stimuli thresholds computed for 70.7% correct responses.

## 2. Study 1: Spatial TOJ

Study 1 was designed to test whether spatial TOJ thresholds measured by the adaptive procedure would differ significantly from spatial TOJ thresholds measured by the method of constant stimuli.

### 2.1. Materials and Methods

#### 2.1.1. Participants

Study 1 included a total of 109 participants, of whom 50 were tested using the adaptive procedure, and 59 were tested using the constant stimuli procedure, in a between-subjects study design. The participants’ ages ranged between 20 and 35, and they were screened for normal hearing (≤20 dB HL at 0.5, 1, 2, and 4 kHz). None of the participants reported any learning disabilities or history of ADHD.

#### 2.1.2. Task, Stimuli, and Procedure

The experiment was approved by the University Ethics Committee. The participants provided signed informed consent before participating in the experiment. All participants were tested on the spatial TOJ paradigm. In this task, participants are required to reproduce the order of two identical tones presented asynchronously to each ear, and to report whether the order of the tones was either right–left or left–right. Both stimuli were 15 ms 1 kHz pure tones with 2 ms cosine-squared rise/fall envelopes, presented at 65 dB SPL. For half of the trials the order of the presentation of the tones was the left ear first followed by the right ear, with the reverse order for the other half of the trials. Participants pressed relevant keyboard keys in the order corresponding to the order of the tones that they heard. Before the experiment began, the participants were trained to recognize the stimuli and perform order judgments (see [4] for a detailed description of the training). 

Each group performed the spatial TOJ task using one of the following psychophysical methods:

*Adaptive procedure*. A 2-down-1-up two-alternative forced choice procedure was used. The initial ISI was 200 ms and was changed according to the participant’s response with the following step sizes: 25 ms for ISI ranging from 100 to 200 ms; 10 ms for ISI ranging between 50 and 100 ms; 5 ms for ISI ranging between 15 and 50 ms; and 2.5 ms for ISI ranging between 0 and 15 ms. The experiment was terminated after 10 reversals, and the threshold was calculated as the average ISI of the last eight presentations (see also [21]).

*Method of constant stimuli*. The tone pairs were presented with an ISI of 5, 10, 15, 30, 60, 90, 120, or 240 ms. Each ISI value was presented randomly and repeated 20 times, resulting in a total of 320 trials (8 ISIs × 2 orders × 20). The TOJ threshold was calculated using the best fit psychometric function and defined as the ISI necessary for 75% or 70.7% correct responses (see also [4,11,16,24]).

The duration of the experiment for all participants, including training, was 20 to 30 min.

### 2.2. Results

Four participants in the Constant Stimuli group did not reach 75% at any of the ISI values. Therefore, they were excluded from the analysis. The threshold distributions of both methods were not found to be significantly different from normal (adaptive procedure: skewness = 0.29, SE = 0.31; method of constant stimuli: skewness = 0.54, SE = 0.35; Figure 1). Levene’s test for homogeneity of variances showed no difference between the methods in the threshold standard deviations (F(1,103) = 0.268, *p* = 0.606). The mean spatial TOJ threshold in the Adaptive Procedure group was 62.85 ms (SD = 29.52 ms) and in the Constant Stimuli group was 63.53 ms (SD = 32.46 ms) (Figure 2). There was no significant difference in the spatial TOJ thresholds between the two groups (t(103) = −0.113, *p* = 0.911). Furthermore, the Bayes factor (BF01 = 6.56) suggested substantial support for H0. This indicates that the data from spatial TOJ are 6.56 times more likely to occur under H0 (no difference between psychophysical methods), than under H1 (significant difference between psychophysical methods). The mean threshold calculated for 70.7% correct responses in the method of constant stimuli was 54.27 ms (SD = 34.8 ms). Although this mean threshold was significantly shorter than that obtained for 75% correct responses (t(46) = −10.070, *p* < 0.001), it was not different from the threshold obtained in the adaptive procedure (t(104) = 1.373, *p* = 0.173).

### 2.3. Discussion

The spatial TOJ thresholds measured by the two psychophysical procedures, the adaptive and constant stimuli methods, did not significantly differ in any of the following measures: (1) the form of the distribution; (2) the mean; or (3) the standard deviation. This lends credence to the possibility of comparing the results of spatial TOJ studies in which different methodologies, i.e., adaptive procedures and the method of constant stimuli, were used.

## 3. Study 2: Spectral TOJ

We found no significant difference in the spatial TOJ thresholds, SDs, or shapes of the threshold distributions generated when using the adaptive procedure versus using the method of constant stimuli. In previous studies, we showed that although both spatial and spectral TOJ are used to measure auditory temporal processing, their response patterns are different [4,26]. Consequently, Study 2 was designed to test whether the psychophysical methods tested in Study 1 affect the measurement of spectral TOJ thresholds differently.

### 3.1. Materials and Methods

#### 3.1.1. Participants

Study 2 included a total of 223 participants, 119 in the Adaptive Procedure group, and 104 in the Constant Stimuli group, in a between-subjects study design. Participants’ ages ranged between 20 and 35, and they were screened for normal hearing (≤ 20 dB HL at 0.5, 1, 2, and 4 kHz). None of the participants reported any learning disabilities or history of ADHD.

#### 3.1.2. Task, Stimuli, and Procedure

The experiment was approved by the University Ethics Committee. The participants provided signed informed consent before participating in the experiment. All participants were tested on the spectral TOJ paradigm. In this task, participants are required to reproduce the order of two pure tones that differ in their pitch and are presented synchronously to both ears. Participants reported whether the order of the tones was high–low or low–high in pitch. The stimuli were 15 ms, 1 kHz, and 1.8 kHz pure tones, with 2 ms cosine-squared rise/fall envelopes, presented at 65 dB SPL. For half of the trials, the order of the presentation of the tones was the high-pitched tone first, followed by the low-pitched tone, and the reverse order was used in the other half of the trials. Participants pressed the relevant keyboard keys in the order corresponding to the order they heard. Before the experiment started, the participants were trained to recognize the stimuli and perform order judgments (see [4] for detailed description of the training). 

Each group performed the spectral TOJ task using one of the psychophysical methods described above and in Study 1.

### 3.2. Results

In both the Adaptive Procedure and Constant Stimuli groups there were participants that did not reach the threshold: in the Adaptive Procedure group, 22 out of the 119 participants (20%) had five consecutive errors at the longest ISI value (240 ms), and in the Constant Stimuli group, 21 out of the 104 participants (20%) did not reach 75% correct responses at any ISI value. Moreover, 58 out of the 119 participants in the Adaptive Procedure group (48.7%) were able to discriminate temporal order at ISI = 0 ms, and 62 out of 104 participants in the Constant Stimuli group (59.6%) reached above 75% correct responses rates at the shortest ISI value (5 ms). The finding that a large number of participants did not reach the threshold criterion (either they were able to judge temporal order at greater than a 75% correct response rate at ISI below 5 ms or they were unable to judge temporal order even when ISI was 240 ms) is consistent with previous reports in the literature [2,3,4,22,26]. Importantly, there was no significant difference in the number of participants that did not reach the threshold (either in very long or very short ISIs) between the Adaptive Procedure and Constant Stimuli groups (χ2(2) = 4.57, *p* = 0.10). In the current study, these participants were excluded from the analysis of assessed thresholds, leaving 39 participants in the Adaptive Procedure group and 21 participants in the Constant Stimuli group whose spectral TOJ thresholds were between 0 ms and 240 ms. 

The spectral TOJ threshold distributions for these 60 participants measured by the two methods were not found to be significantly different from normal (adaptive procedure: skewness = 0.73, SE = 0.38; method of constant stimuli: skewness = 0.51, SE = 0.50; Figure 3). Although more participants in the Constant Stimuli group had very short thresholds (0–20 ms), and more participants in the Adaptive Procedure group had very long thresholds (140 ms and above), it was not statistically significant (χ^2^_(10)_ = 7.820, *p* = 0.646). Moreover, Levene’s test for homogeneity of variances showed no difference between the standard deviations of the two threshold distributions (F(1,58) = 2.695, *p* = 0.106). The mean spatial TOJ threshold in the Adaptive Procedure group was 81.64 ms (SD = 52.64 ms), and in the Constant Stimuli group it was 59.51 ms (SD = 37.21 ms) (Figure 4). However, a *t*-test showed that this difference in mean thresholds did not reach statistical significance (t(58) = 1.891, *p* = 0.064). The Bayes factor (BF01 = 1.35) suggested anecdotal support for H0. This indicated that the data from spectral TOJ were only 1.35 times more likely to occur under H0 (no difference between psychophysical methods), than under H1 (significant difference between psychophysical methods). The mean threshold calculated for 70.7% correct responses for the method of constant stimuli was 55.39 ms (SD = 38.83 ms). It was not different from the mean threshold calculated for 75% correct responses (t(58) = −0.558, *p* = 0.597), nor from mean threshold of the adaptive procedure (t(58) = 1.513, *p* = 0.137).

### 3.3. Discussion

Although visual inspection of spectral TOJ mean thresholds and threshold distribution shows they might differ between the adaptive procedure and the method of constant stimuli, statistical analysis showed that they did not differ in (1) the form of the distribution; (2) the mean; or (3) the standard deviation. However, since the Bayes factor here was small, the conclusion that there is no difference between the psychoacoustic methods tested should be taken cautiously. It should be noted that a large number of participants were excluded from the analysis, and that the analysis reflects only those whose thresholds were between 0 and 240 ms and thus met the threshold criterion. Including only these participants resulted in normal distributions for spectral TOJ thresholds, unlike the “bowl” shaped distribution reported several times previously, when all participants have been included [21,26]. However, that phenomenon still remains with no solution as to its cause and potential solution. In addition, although there was no difference in the threshold variance between the methods, both groups’ variance was large. This could have affected the comparison between the methods. These limitations call for further research into the nature of spectral TOJ to examine the causes that prevent some of the participants from having thresholds, and the large variances in thresholds for those who achieved thresholds. In conclusion, although analysis of spectral TOJ data might be problematic, we have no evidence that either of the methodologies can result in shorter thresholds for participants whose spectral TOJ thresholds are larger than 0 ms and shorter than 240 ms.

## 4. General Discussion

The comparison between the adaptive procedure and the method of constant stimuli provided similar results for both spatial and spectral TOJ. The data obtained for spatial TOJ provided strong evidence that there is no significant difference between the thresholds measured by either of the two methods. Consequently, one may safely compare the results of studies of spatial TOJ when one study used an adaptive procedure while the other used the method of constant stimuli. Although the results of the study comparing spectral TOJ thresholds obtained by an adaptive procedure with those obtained by the method of constant stimuli were similar (no difference), the conclusion is less clear. This might be due to the smaller number of participants analyzed in the spectral TOJ (a total of 60 participants, versus a total of 105 in spatial TOJ), although the initial pool of participants was much larger (223 participants, versus 109 in spatial TOJ). 

### 4.1. Limitations and Future Studies

The present study was performed as a between-subjects design, although a within-subjects design might have better compensated for the variance observed in the data (especially in the spectral TOJ). This poses a limitation for the study, but was chosen for two reasons: (1) previous studies showed that spectral TOJ has a high learning effect; therefore, we did not want participants to repeat it twice, each with a different method; and (2) having similar thresholds with different groups of participants makes a stronger argument for the similarity between methods than it would with the same participants. Moreover, one should take into consideration that thresholds obtained from the adaptive procedure reflect different threshold criteria to those of the method of constant stimuli, and therefore have different accuracy rates. In the present study, we mainly focused on data similar to those found in the literature (i.e., thresholds for 75% correct responses for the method of constant stimuli), but we also provided a comparison to data more directly comparable to those obtained from the adaptive procedure (i.e., 70.7% correct response rates). Future studies should take into account that no difference was found between the methods, regardless of the threshold criteria.

### 4.2. Conclusions

When answering the question towards which the study was directed: is there any difference in the thresholds determined by using an adaptive procedure versus the thresholds determined by the method of constant stimuli? The initial answer provided by the present study was complex. The answer we found was that for spatial TOJ it did not matter, since either method provided the same estimate of the TOJ threshold. Therefore, the present study’s contributions are (1) to facilitate comparisons between studies of spatial TOJ using these methods, and (2) to facilitate the use of datasets derived from both methods to answer research questions. For spectral TOJ, the answer was more cautious. On the one hand, there were no differences between the threshold distributions and mean thresholds. On the other hand, there was a large variance in the thresholds determined by each method, and there was a large number of participants that did not reach the criterion for the threshold, and consequently were not part of the analysis. These facts may have contributed to the finding of there being no substantial difference in the thresholds determined by the two different methodologies. Thus, any conclusions regarding spectral TOJ in the current study, and for other studies, might be problematic.

## Figures and Tables

**Figure 1 sensors-22-04830-f001:**
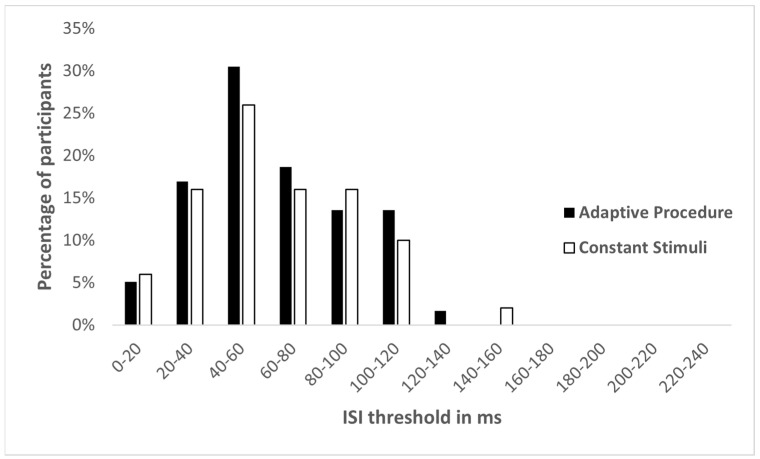
Threshold distribution for spatial TOJ in adaptive procedure and method of constant stimuli.

**Figure 2 sensors-22-04830-f002:**
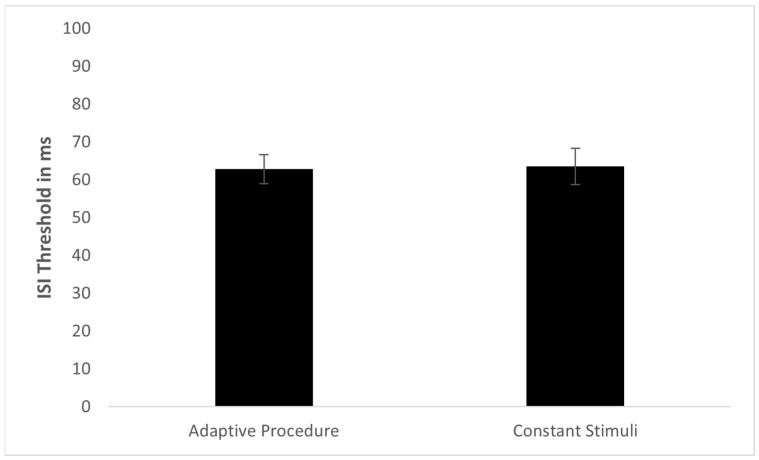
Mean and standard error for spatial TOJ thresholds in adaptive procedure and method of constant stimuli.

**Figure 3 sensors-22-04830-f003:**
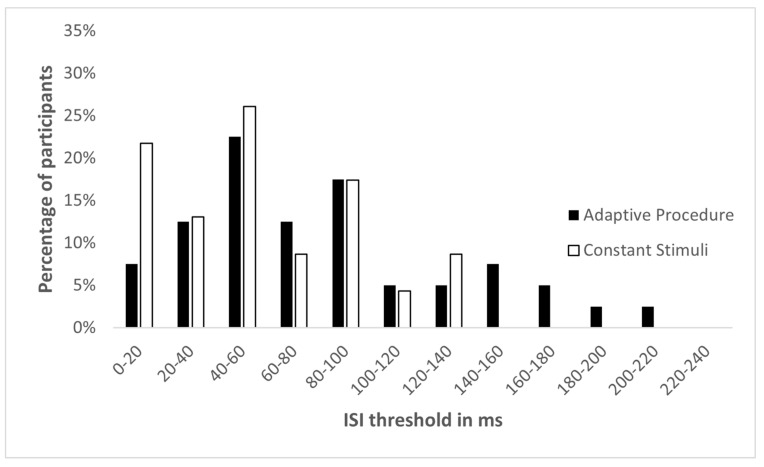
Threshold distribution for spectral TOJ in adaptive procedure and method of constant stimuli.

**Figure 4 sensors-22-04830-f004:**
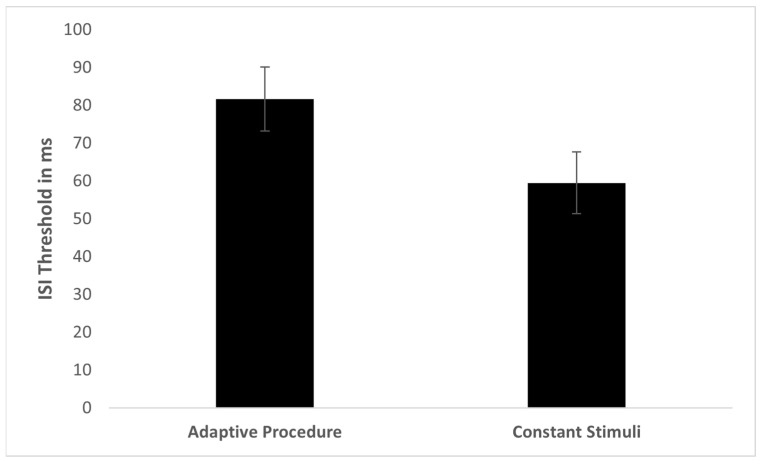
Mean and standard error for spectral TOJ thresholds in adaptive procedure and method of constant stimuli.

**Table 1 sensors-22-04830-t001:** Spatial and spectral TOJ thresholds by psychophysical methods.

**a.** Spatial TOJ.			
**Study**	**Stimuli**	**n**	**ISI Threshold (ms)**
*Adaptive Procedure*			
Fink et al. (2005) [2]	1 ms clicks	20	54
Fink et al. (2005) [2]	1 ms clicks	20	52
Fink et al. (2006) [3]	1 ms clicks	49	57
Fostick & babkoff (2013) [4]	15 ms 1 kHz tones	30	65
Fostick et al. (2014) [11]	15 ms 1 kHz tones	47	58
Fostick et al. (2014) [25]	15 ms 1 kHz tones	20	67
Fostick & Babkoff (2017) [26]	15 ms; 1 kHz tones	37	72
Kinsbourne et al. (1991) [27]	1 ms clicks	21	47
Lotze et al. (1999) [28]	1 ms clicks	5	37
Lotze et al. (1999) [28]	1 ms clicks	5	21
Lotze et al. (1999) [28]	1 ms clicks	2	24
Szymaszek et al. (2006, 2009) [17,18]	1 ms clicks	17	68
*Constant Stimuli* ^1^			
Babkoff & Fostick (2013) [1]	5 ms 1 kHz tones	28	114
Babkoff & Fostick (2013) [1]	10 ms 1 kHz tones	28	97
Babkoff & Fostick (2013) [1]	20 ms 1 kHz tones	28	79
Babkoff & Fostick (2013) [1]	30 ms 1 kHz tones	28	57
Babkoff & Fostick (2013) [1]	40 ms 1 kHz tones	28	42
Babkoff et al. (2005) [19]	10 ms 1 and 1.5 kHz tones	18	63
Ben-Artzi et al. (2005) [8]	15 ms 300 Hz and 600 Hz tones	26	49
Fostick et al. (2012) [9]	15 ms 1 kHz tones	40	68
Fostick et al. (2012) [10]	15 ms 1 kHz tones	46	74
Fosticket al. (2014) [25]	15 ms 1 kHz tones	18	50
Fostick & Babkoff (2017) [26]	15 ms; 0.3/0.6/1 kHz tones	185	80
Kolodziejczyk & Szelag (2008) [29]	15 ms 300 Hz tones	17	37
**b.** Spectral TOJ.			
**Study**	**Stimuli**	**n**	**ISI Threshold (ms)**
*Adaptive Procedure*			
Fink et al. (2005) [2]	10 ms; 800 Hz & 1.2 kHz	20	15
Fink et al. (2005) [2]	10 ms; 800 Hz & 1.2 kHz	20	20
Fink et al. (2006) [3]	10 ms; 600 Hz & 1.2 kHz	45	21
Fostick & Babkoff (2013) [4] Fostick et al. (2014) [11]	15 ms; 1 & 1.8 kHz	19	45
Fostick & Babkoff (2017) [26]	15 ms; 1 & 1.1/1.5/1.8/3.5 kHz	99	88
Stevens & Weaver (2005) [30]	20 ms; 1 & 4 kHz	11	19
Szymaszek et al. (2006, 2009) [17,18]	10 ms; 400 Hz & 3 kHz	17	33
*Constant Stimuli* ^1^			
Fostick et al. (2008) [31]	15 ms; 1 & 1.8 kHz	50	2
Fostick et al. (2012) [9]	15 ms; 1 & 1.8 kHz	40	46
Fostick & Babkoff (2017) [26]	5–20 ms; 1 & 1.5/1.8 kHz	289	75
Kanabus et al. (2002) [32]	15 ms; 300 Hz & 3 kHz	12	40

^1^ Threshold criteria at 75%.

## Data Availability

The data presented in this study are openly available at: https://www.kaggle.com/datasets/leahfostick/psychoacoustic-methods-for-toj (accessed on 22 May 2022).

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
