# Peer review of "The Effects of Psychophysical Methods on Spectral and Spatial TOJ Thresholds"

_sensors, 2022, doi:10.3390/s22134830_

Round 1

Reviewer 1 Report

This paper finds no significant difference between the two methods of measuring spatial and spectral OJ thresholds. The presentation is clear and the results are meaningful. I have the following comments:

1. Line 18: change "their results" to "these results"

2. Line 19: "similar result" to " A similar result"

3. Line 63: "some compare" to "some studies compared"

4. Lines 169 to 184: It is surprising that such a large number of subjects did not reach the thresholds. More discussions are needed to explain the cause and potential solutions in future studies. 

5. Lines 190-192: The thresholds are quite different but the statistical analysis showed no significant difference. Please explain further.

6. Line 200: Fig. 3 shows a big difference for the 0-20ms condition. Please explain further.

Author Response

Reviewer 1

  1. Line 18: change "their results" to "these results"

Response: This sentence refers to the possibility to compare the results of the adaptive procedure and the results of the method of constant stimuli. Therefore, the sentence refers to “their results”.

  1. Line 19: "similar result" to " A similar result"

Response: Corrected.

  1. Line 63: "some compare" to "some studies compared"

Response: Corrected.

  1. Lines 169 to 184: It is surprising that such a large number of subjects did not reach the thresholds. More discussions are needed to explain the cause and potential solutions in future studies. 

Response: Unfortunately, although several studies report that spectral TOJ thresholds cannot be computed to groups of participants, none suggested the cause and potential solutions for it. we this limitation to the Experiment 2 Discussion section, along with a call to further investigate this phenomenon.

On page 7, line 234, we note:

“It should be noted, that large number of participants were excluded from the analysis, and the analysis reflects only those whose thresholds were between 0 and 240 ms and thus met the threshold criterion. Including only these participants resulted in normal distributions for spectral TOJ thresholds, unlike the “bowl” shaped distribution re-ported several times previously, when all participants are included [21,26]. However, that still remains with no solution as to its cause and potential solution. Also, although there was no difference between methods in thresholds variance, both groups’ variance was large. This could have affected the comparison between methods. These limitations call for further research into the nature of the spectral TOJ to test what are the causes that prevent some of the participants from having thresholds, and the large variances in thresholds for those who achieved thresholds.”

  1. Lines 190-192: The thresholds are quite different but the statistical analysis showed no significant difference. Please explain further.

Response: We agree that the mean thresholds are quite different, although this difference did not reach statistical significance. It might be due to a large variance in thresholds produced by both methods (no difference in variance between the methods). This explanation was added to the Experiment 2 Discussion section.

On page 7, line 240, we note:

“Also, although there was no difference between methods in thresholds variance, both groups’ variance was large. This could have affected the comparison between methods. These limitations call for further research into the nature of the spectral TOJ to test what are the causes that prevent some of the participants from having thresholds, and the large variances in thresholds for those who achieved thresholds.”

  1. Line 200: Fig. 3 shows a big difference for the 0-20ms condition. Please explain further.

Response: In light of this comment, we added a comparison of the threshold prevalence between the Constant Stimuli Adaptive Procedure groups, but this comparison was not statistically significant (χ2(10) = 7.820, p = .646). This was added to the Experiment 2 Results section.

On page 6, line 205, we note:

“Although more participants in the Method of Constant Stimuli group had very short thresholds (0-20 ms), and more participants in the Adaptive Procedure group had very long thresholds (140 ms and above), it was not statistically significant (χ2(10) = 7.820, p = .646).”

Reviewer 2 Report

I thank MPDI to offer me an opportunity to review this paper. 

In this work, the authors made comparison of an adaptive procedure, and the method of constant stimuli to evaluate the temporal-order judgment (TOJ) tasks.

Two Studies are performed with different participants: study 1 included a total of 109 participants (50 vs. 59) and study 2 included a total of 223 participants (119 vs. 104) in the adaptive procedure group, and in the Constant Stimuli group, respectively. 

The authors conclude that no difference between spatial TOJ thresholds measured by an adaptive procedure and the method of constant stimuli.

I think it is an accessible paper as the authors delivered a nice way to communicate their findings to readers.

In addition, from statistical perspective, I think the authors describe the statistical analysis results clearly.

Below I have a few minor concerns that hope the authors can addressed them.

**point** It would be good if authors can make more discussion how this work would be interesting to many researchers, and how that their work will likely stimulate further research in the field.

**point** Table 1a and Table 1b seems to be the same. I cannot tell the difference between the two in the list references. 

**point** As claimed in the MDPI, I am hoping the authors can provide their data and script that reproduces the results in the papers(the histogram, the barplots, and the test statistics, in particular how do the Bayes factors are obtained from their analysis).

**point** Figures 1, 3: it would be  also good to report the Kolmogorov-Smirnov test statistics (based on comparing two cumulative distribution functions)  to compare the two empirical data distributions, as well as to compare one empirical data distribution to the normal distribution.

Author Response

Reviewer 2

**point** It would be good if authors can make more discussion how this work would be interesting to many researchers, and how that their work will likely stimulate further research in the field.

Response: In light of this comment, a “conclusions” section was added to the General Discussion, within it the present study’s contributions are (1) to facilitate comparisons between studies of spatial TOJ using these methods and (2) to facilitate the use of datasets derived from both methods to answer research questions.

On page 8, line 282, we note:

“Therefore, the present study’s contributions are (1) to facilitate comparisons between studies of spatial TOJ using these methods and (2) to facilitate the use of datasets de-rived from both methods to answer research questions.”

**point** Table 1a and Table 1b seems to be the same. I cannot tell the difference between the two in the list references. 

Response: This mistake was corrected.

**point** As claimed in the MDPI, I am hoping the authors can provide their data and script that reproduces the results in the papers(the histogram, the barplots, and the test statistics, in particular how do the Bayes factors are obtained from their analysis).

Response: The materials of the present study are openly available at: https://www.kaggle.com/datasets/leahfostick/psychoacoustic-methods-for-toj

Reviewer 3 Report

This paper examines whether the threshold of temporal order judgment (TOJ) is affected by the measurement method. Two measurement methods are discussed: the constant method and the adaptive method, which have been employed in many studies. Experiments with a large number of listeners are conducted for both spatial and spectral TOJ, and it is concluded that the distributions of the threshold for both types of TOJ are not affected by the measurement method.

I evaluate this paper as Reject based on the following two issues.

(1) The motivation for the present study is unclear. 

As far as I know,  in most cases, there is no significant difference between the two methods in terms of threshold measurement accuracy, although the adaptive method introduces some biases as the author describes. The motivation for the present study is unclear because it is not stated what specific problems are expected in using the adaptive (or constant) method for auditory TOJ threshold measurement. 

(2) Improper experimental design.

As the results of this experiment and those of Szelag et al. (2018) show, the threshold of spectral TOJ varies greatly among individuals. Nevertheless, the present experiment employs a between-subjects design, which makes a fair comparison of the two measurement methods difficult.

Also, the following is a minor point of concern.

The probability of convergence of the 2-down-1-up AFC method is 70.7%. If the purpose is to compare measurement methods within the present study rather than to compare with previous studies, it seems that 70.7% should also be used when obtaining the threshold from the constant-stimuli method. Alternatively, if not the threshold but the psychometric function is compared between the two measurement methods, as in Dai (1995), for example, it seems that the selection of listeners in the analysis of the spectral TOJ is no longer necessary.

Szelag et al.(2018): https://doi.org/10.3389/fpsyg.2018.02557

Dai (1995): https://doi.org/10.1121/1.413802

Author Response

Reviewer 3

(1) The motivation for the present study is unclear.

As far as I know, in most cases, there is no significant difference between the two methods in terms of threshold measurement accuracy, although the adaptive method introduces some biases as the author describes. The motivation for the present study is unclear because it is not stated what specific problems are expected in using the adaptive (or constant) method for auditory TOJ threshold measurement.

Response: Indeed, there are studies that compared psychophysical methods in different psychophysical tasks, in an attempt to learn which is the best method to reflect the individual’s ability, or which is the most effective to obtain the most accurate measure. However, no study to date tested whether Adaptive Procedure and the Method of Constant Stimuli produce the same thresholds in spectral and spatial TOJ. Thus, it is not clear whether it is possible to draw conclusions based on data collected in both methods. Also, in the era of big data based on analyses from different datasets, it is important to know whether thresholds collected in different methods can be combined together. This rationale was added to the Introduction.

On page 6, line 63, we note:

“Which psychophysical method is more valid or more reliable for determining thresholds? Several studies have addressed this question in different psychophysical tasks. Some studies compared the thresholds obtained by each method in an attempt to discover which method reflects the individual’s ability better [30,31], while others focused on their effectiveness in an attempt to obtain the most accurate measure that requires the least time and effort by the participants [32,33]. However, no study to date tested whether Adaptive Procedure and the Method of Constant Stimuli produce the same thresholds in spectral and spatial TOJ. Thus, it is not clear whether it is possible to draw conclusions based on data collected in both methods. Also, in the era of big data based on analyses from different datasets, it is important to know whether thresholds collected in different methods can be combined together.”

(2) Improper experimental design.

As the results of this experiment and those of Szelag et al. (2018) show, the threshold of spectral TOJ varies greatly among individuals. Nevertheless, the present experiment employs a between-subjects design, which makes a fair comparison of the two measurement methods difficult.

Response: We agree with the reviewer that a with-subjects design is better for a task with large variance. Nevertheless, we chose a between-subjects design for the present study, for two reasons: (1) previous studies showed that spectral TOJ has a high learning effect. Therefore, we did not want participants to repeat it twice, each with a different method; and (2) having similar thresholds with different groups of participants makes a stronger argument for the similarity between methods than with the same participants. In light of this and the following comment, we added a “Limitations and future studies” section to the General discussion and added this issue to this section.

On page 8, line 262, we note:

“The present study was performed in a between-subjects design, although a with-in-subjects design might have better compensated for the variance observed in the data (especially in the spectral TOJ). This poses a limitation for the study but was chosen due to two reasons: (1) previous studies showed that spectral TOJ has a high learning effect. Therefore, we did not want participants to repeat it twice, each with a different method; and (2) having similar thresholds with different groups of participants makes a stronger argument for the similarity between methods than with the same participants. Also, one should take into consideration that thresholds obtained from Adaptive Procedure to those of the Method of Constant Stimuli reflect different threshold criteria, and there-fore different accuracy rates.”

Also, the following is a minor point of concern.

The probability of convergence of the 2-down-1-up AFC method is 70.7%. If the purpose is to compare measurement methods within the present study rather than to compare with previous studies, it seems that 70.7% should also be used when obtaining the threshold from the constant-stimuli method. Alternatively, if not the threshold but the psychometric function is compared between the two measurement methods, as in Dai (1995), for example, it seems that the selection of listeners in the analysis of the spectral TOJ is no longer necessary.

Response: Since the aim of the current study was to facilitate comparisons between data reported in the literature, the main analyses used these threshold measures (and thus, comparisons between thresholds obtained for 75% to those obtained for 70.7%). Nevertheless, in light of this comment, we added a direct comparison of the adaptive thresholds to constant stimuli thresholds computed for 70.7% correct, for both spatial and spectral TOJ. In both tasks, no differences were found between adaptive TOJ thresholds and constant stimuli thresholds computed for 70.7%. This topic was added to the Introduction, to Experiments 1 and 2 Results section, and to the Limitation section of the General Discussion.

On page 3, line 77, we note:

“It should be noted that thresholds for the method of constant stimuli are calculated as the ISI for 75% correct, while thresholds in the adaptive procedure are equivalent to 70.7%. Since the aim of the current study was to facilitate comparisons between data reported in the literature, the main analyses used these threshold measures (and thus, comparisons between thresholds obtained for 75% to those obtained for 70.7%). Nev-ertheless, we added also a direct comparison of the adaptive thresholds to constant stimuli thresholds computed for 70.7% correct.”

On page 4, line 137, we note:

“The mean threshold calculated for 70.7% in the Method of Constant Stimuli was 54.27 ms (SD = 34.8 ms). Although this mean threshold was significantly shorter than that obtained for 75% (t(46) = -10.070, p < .001), it was not different from the threshold ob-tained in the Adaptive Procedure (t(104) = 1.373, p = .173).”

On page 6, line 218, we note:

“The mean threshold calculated for 70.7% in the Method of Constant Stimuli was 55.39 ms (SD = 38.83 ms). It wasn’t different from the mean threshold calculated for 75% (t(58) = -.558, p = .597) nor from mean threshold of the Adaptive Procedure (t(58) = 1.513, p = .137).”

On page 8, line 271, we note:

“In the present study, we mainly focused on the figures similar to those are going to be found in the literature (i.e., thresholds for 75% correct in the Method of Constant Stimuli), but also provided a comparison to figures more directly comparable to those of the Adaptive Procedure (i.e., 70.7% correct). Future studies should take into account that no difference between methods was found regardless of threshold criteria.”

Round 2

Reviewer 3 Report

Based on the revised manuscript by the authors, I change my evaluation from Reject to Accept. The following are comments on future work and the manuscript does not require further revisions.

Regarding the issue of experimental design, two responses were made by the authors. Regarding (1), if the focus is on the learning effect in the comparison of the two measurement methods, as the author said, a within-subjects design would be difficult. On the other hand, if the focus is on the threshold after the learning effect has reached a steady state, I think a within-subjects design can be used by truncating the first few sessions. Regarding (2), even if we obtain the result that "the thresholds obtained from two different groups are not different," I think we need to be careful because it is possible that the thresholds of the two groups were potentially different but were offset by the difference between measurement methods, resulting in equal threshold values.